# Pore Structure and Properties of PEEK Hollow Fiber Membranes: Influence of the Phase Structure Evolution of PEEK/PEI Composite

**DOI:** 10.3390/polym11091398

**Published:** 2019-08-26

**Authors:** Gong Chen, Yuan Chen, Tingjian Huang, Zhongchen He, Jianjun Xu, Pengqing Liu

**Affiliations:** College of Polymer Science & Engineering, Sichuan University, Chengdu 610065, China

**Keywords:** poly(ether ether ketone), hollow fiber membrane, phase structure, pore characteristics

## Abstract

Poly(ether ether ketone) (PEEK) hollow fiber membranes were successfully prepared from miscible blends of PEEK and polyetherimide (PEI) via thermally-induced phase separation (TIPS) with subsequent extraction of the PEI diluent. The phase structure evolution, extraction kinetics, membrane morphology, pore size distribution and permeability for the hollow fiber membrane were studied in detail. Extraction experiments, differential scanning calorimetry (DSC) and dynamic mechanical thermal analysis (DMA) studies showed that the heat treatment had a significant influence on the two-phase structure of PEEK/PEI, and that it was controlled by the crystallization kinetic of PEEK and the diffusion kinetic of PEI. As the annealing temperature increased, the controlling factor of the phase separation changed from PEEK crystallization to PEI diffusion, and the main distribution of the amorphous PEI chains were changed from the interlamellar region to the interfibrillar or interspherulitic regions of PEEK crystallization. When the annealing temperature increased from 240 °C to 280 °C, the extracted amount of PEI increased from 85.19 to 96.24 wt %, and the pore diameter of PEEK membrane increased from 10.59 to 37.85 nm, while the surface area of the PEEK membrane decreased from 111.9 to 83.69 m^2^/g. Moreover, the water flux of the PEEK hollow fiber membranes increased from 1.91 × 10^−2^ to 1.65 × 10^−1^ L h^−1^ m^−2^ bar^−1^ as the annealing temperature increased from 240 °C to 270 °C. The structure and properties of the PEEK hollow fiber membrane can be effectively controlled by regulating heat treatment conditions.

## 1. Introduction

Membranes are significantly prevalent because fluid components avoid phase change during most membrane separations, and save energy when compared with traditional distillation processes. Their applications in almost all fields of research and industrial activities are widely consolidated [1,2,3,4]. Inorganic membranes are witnessing a shortage, owing to their hydrolytic stability, poor preparation and high cost, therefore, polymer membranes have become a hot research topic [5,6,7]. At present, there are many polymer membranes under study, such as the polyamide membrane, polysulfone membrane and polyolefin membrane [8,9,10]. However, there is a common problem in these membrane materials that causes function failure in relatively high temperatures and in corrosive liquid, which limits their application in the field of membrane separation [11,12]. Therefore, it is necessary to find a high performance membrane material that can overcome the shortcomings of existing materials and broaden the application field of membrane separation technology. Poly(ether ether ketone) (PEEK) is a wholly aromatic, semi-crystalline thermoplastic engineering plastic with excellent dimensional stability and mechanical properties, and it can be used for a relatively long time under the temperature of 250 °C. The excellent resistance to chemical corrosion of PEEK makes it an ideal material for preparing high temperature-resistant and chemical-resistant membranes [13,14,15].

A large amount of research about the porous PEEK preparation via the thermally-induced phase separation (TIPS) method has been conducted, and the choice of porogenic diluent is the factor of greatest importance [16,17]. In earlier studies on this method, porous PEEK membranes were prepared by using high boiling solvents and plasticizers, which are capable of dissolving the PEEK at relatively high temperatures, resulting in low solution viscosity and solvent evaporation at such operating temperatures, making the membrane formation more difficult than ever imagined [18,19]. Some effort has been made to overcome these aforementioned difficulties to prepare porous PEEK membranes from the wet form, namely to find a polymeric diluent. Poly(ether imide) (PEI) came into view as an ideal porogenic diluents, because PEEK and PEI are wholly miscible when heated into melt, and PEI is soluble in many solvents while PEEK is stable at such a condition [20,21,22].

PEEK is a semi-crystalline polymer, while PEI is an amorphous polymer. In the process of membrane formation, crystalline polymers undergo crystallization while amorphous polymers remain unchanged [22,23]. The crystalline solidification of homogeneously mixed polymer blends is a highly complicated process, and phase separation occurs during crystallization of the PEEK component. Among others, both the kinetic competition between the proceeding of crystallization of the crystalline polymer component and the chain diffusion ability of the amorphous polymer play significantly essential roles. This is related to the evolution of the phase structure of the blended polymers, which has a significant influence on the pore structure, morphology and membrane properties of the final membrane [24]. The solid–liquid phase separation characteristics and phase structure of the PEEK/PEI blends have been investigated by microscopy and thermal analysis methods [24,25]. Hudson et al. firstly studied the phase structure of PEEK/PEI, and the results showed that PEI, acting as a diluent, was rejected from growing PEEK crystals, forming three possible distribution states, namely, interlamellar, interfibrillar and interspherulitic. It was found that PEI was in the interlamellar morphologies at a low crystallization temperature and PEI was in the structure of interfibrillar or interspherulitic morphologies when the crystallization temperature was high [26]. The effect of annealing temperature on the phase structure of PEEK/PEI was investigated by Mehta et al., and showed that the interfibrillar or interspherulitic phase might dominate at the high annealing temperature, and extraction from interspherulitic pockets produced relatively coarse pores [27].

The PEEK/PEI hollow fiber membrane was successfully prepared by the researchers in the PoroGen company from the United States using the TIPS method, focusing on the feasibility of the preparation of a PEEK hollow fiber membrane; however, the effects of phase structure evolution on the structure and properties of the aforementioned hollow fiber membrane were not taken into account [28]. From the previous studies, the evolution of PEEK/PEI phase structure might have an important influence on the structure and properties of the PEEK porous membrane.

In this present work, PEEK hollow fiber membrane was prepared from miscible blends of PEEK and PEI via the TIPS method. The influence of the evolution of the aggregation structure during the process on the phase separation and phase structure was studied in depth, and the relationships among the process conditions, membrane structure and membrane properties were clarified. Finally, it is hoped that the structure and properties of the PEEK hollow fiber membrane can be effectively controlled.

## 2. Experimental

### 2.1. Materials

The PEEK powder (330G) was purchased from Zhong Yan High Performance Engineering Plastics Co., Ltd. (Changchun, China). The PEI resin powder (Ultem 1000) was purchased from SABIC innovative plastics (Riyadh, Saudi Arabia). N-methyl pyrrolidone (NMP), ethanolamine and isopropyl alcohol were purchased from the Kelong Chemical Reagent Factory (Chengdu, China). Deionized water was self-made in the laboratory.

### 2.2. Melt Spinning of the Precursory PEEK/PEI Blend Hollow Fibers

PEEK/PEI particles with 40 wt % of PEEK and 60 wt % of PEI weight fraction were prepared by melting extrusion with a twin-screw extruder (SLJ-30, Longchang Chemical Machinery Factory, Neijiang, China) at 360 °C. Melt spinning of the precursory PEEK/PEI blend hollow fibers were conveniently carried out utilizing a high-temperature extruder with melt spinning modules. The schematic diagram of the spinning device is shown in Figure 1. The hollow fibers were spun and quenched in a water bath in which the temperature was controlled at 25 °C by a constant temperature circulating device, and then collected by the take-up device.

### 2.3. Preparation of PEEK Hollow Fiber Membrane

Annealing of the hollow fiber was carried out for 3 h at a certain temperature, such as 240, 250, 260, 270 or 280 °C. The obtained opaque precursory fibers were then submerged into mixed extraction solution containing 80 vol % N-methyl pyrrolidone (NMP) and 10 vol % monoethanolamine (MEA) and 10 vol % water at 120 °C for 9 h to perform decomposition and the removal of PEI from the precursor fibers. The porous PEEK hollow fiber membranes were obtained by washing with isopropyl alcohol (IPA) and water, and were air dried overnight. The extraction effect was evaluated by a mass loss rate of PEI, which was determined by the mass change rate of PEI before and after extraction:
α = (m_0_ − m_t_)/(m_0_ × 60%) × 100%(1)
where α is the mass loss rate of PEI and m_0_ and m_t_ are the weights of hollow fibers before and extraction, respectively.

The optimal extraction time was determined by the relationship between the mass loss rate and extraction time, as seen in Appendix A, and the decomposition mechanism of PEI in the extraction solution is shown in Appendix A.

### 2.4. Characterization of the Hollow Fiber Membrane

Differential scanning calorimetry (DSC) was carried out using DSC 204F1 (Netzsch, Selb, Bavaria, Germany) at a heating rate of 10 °C/min in the flowing nitrogen, and the testing temperature range was set from 100 to 360 °C. The temperature and heat enthalpy were calibrated with indium standard before the equipment began to run.

Dynamic mechanical thermal analysis (DMA) was carried out in the vicinity of the glass–rubber relaxation operating in uniaxial tension mode using Q800 (TA Instruments, Newcastle, DE, USA). The storage modulus and loss tangent (tan δ) were recorded at a frequency of 1 Hz across a temperature range of 120 to 280 °C; a scanning rate of 0.8 °C/min was used.

The membrane morphology was studied by scanning electron microscope (SEM) observations of the cross-section and inner and outer surfaces (JSM-5900LV, JEOL, Ltd., Tokyo, Japan). Hollow fiber (HF) samples were freeze-fractured in liquid N2 to produce a clean brittle fracture and were subsequently sputter coated with gold before SEM observation.

The pore characteristics were studied by Brunner-Emmet-Teller (BET) measurements (Belsorp-Max, MicrotracBEL, Inc., Osaka, Japan). Average pore size and surface area were measured by the nitrogen absorption-desorption method.

The porosity was calculated according to the commonly used method based on density measurements [29]. The effect of the residual PEI on the porosity of PEEK membrane was ignored, as the densities of PEEK and PEI are very similar.
ε(%) = (1 − ρ_fiber_/ρ_PEEK_) × 100%(2)
where ρ_fiber_ and ρ_PEEK_ are the fiber and PEEK density, respectively. The fiber density was calculated from the mass and volume ratio:
ρ_fiber_ = m/((S_outer_ − S_inner_)·L)(3)
where L, m, S_outer_ and S_inner_ are the length, the mass, the outer and the inner surface area of the fiber, respectively. The known density of the PEEK was 1.30 g cm^−3^.

According to the HY/T 051-1999 standard, water permeability is determined by feeding distilled water to the module and measuring the permeate volume collected in a specific time t at a certain trans-membrane pressure (P):
J = V/(S·t·P)(4)
where S is the internal surface area, since the internal surface is usually denser and it is supposed to have the main resistance. The test device diagram for permeability is shown in Figure 2.

## 3. Results and Discussion

### 3.1. Phase Structure of PEEK and PEI

The solid-liquid phase separation of PEEK/PEI occurs during the crystallization of PEEK, and different annealing treatments cause the changes of the crystallization kinetics of PEEK, possibly resulting in different phase structures. The range of isothermally annealing temperature (240 to 280 °C) was established so that complete crystallization could be obtained at an appropriate crystallization time (i.e., 3 h) from the blend composition. Dynamic mechanical thermal analysis (DMA) is a sensitive method to investigate the glass-rubber relaxation characteristics of the polymer blends. Therefore, DMA can be used to illustrate the nature of PEI phase in the corresponding PEEK/PEI blend. Representative loss tangent data (tan δ) of PEI are plotted versus temperature, as shown in Figure 3a. The result of two peaks in tan δ of PEEK/PEI blend reflects two separated glass transition events for the amorphous phase. The position of the high-temperature relaxation is independent of blend composition and coincides with the relaxation of PEI homopolymer, thus indicating the existence of an essentially pure PEI phase dispersed in the interfibrillar regions of PEEK, together with the amorphous PEEK, and the low-temperature relaxation in tan δ that corresponds to a blend of PEEK and PEI trapped between crystalline lamellae [27]. It can be seen from Figure 3a that the tan δ in the 200 °C of PEEK/PEI 240 °C is higher than that of PEEK/PEI 280 °C. This reflects PEI enrichment in the interlamellar regions, and thus corresponds to greater PEI retention for the annealing temperature of 240 °C blend as compared to 280 °C. At the same time, DSC heating curves (Figure 3b) were recorded to determine the crystalline melting enthalpy for the isothermal annealing blend samples. It is obvious that the melting peak of PEEK/PEI blends is wider than that of neat PEEK. Moreover, a shoulder peak appears near the melting peak, which probably results from the amorphous PEI existing among the PEEK crystal. The crystal structure of PEEK becomes loose, and the melting peak turns broader in the presence of a shoulder peak.

Extraction experiments were carried out using the aforementioned annealed PEEK/PEI composite hollow fibers. In order to remove PEI completely, the PEEK/PEI hollow fiber membrane was extracted for 9 h at 120 °C in the extraction solution. The relationship between mass loss rate of PEI and annealing temperature is shown in Figure 4. Quantitative analysis of Figure 4 shows that PEI could not completely be removed by the reactive extractant. Meanwhile, with the increase of annealing temperature from 240 to 280 °C, the extraction amount of PEI increased from 85.19 wt % to 95.24 wt %. This may have been related to the distribution of PEI in the PEEK crystal. At a high annealing temperature, the phase separation of PEEK/PEI composite may be controlled by the diffusion of the PEI chain, rather than the crystallization of PEEK and PEI chains that are trapped primarily in interspherulitic regions, which is of benefit for the extraction of PEI.

From extraction experiments above, it was found that a small amount of PEI cannot be extracted, and the retained PEI can be determined by DMA curves before and after extraction. As shown in Figure 5a, the disappearance of the relaxation peak at 230 °C associated with interfibrillar/interspherulite segregation after extraction indicated that the PEI chains trapped in interfibrillar and interspherulite regions were completely extracted. However, the relaxation peak at a low temperature, corresponding to the miscible part of PEEK and PEI, did not completely disappear, and the position shifted to the lower temperature. This result indicated that the PEI trapped in the interlamellar regions was retained during extraction, which also verified the results of the previous extraction experiments. It is also illustrated in Figure 5b that the melting peak of PEEK hollow fiber membranes after extraction was similar to that before extraction. From the previous extraction experiments, PEI is not removed completely, which is the main reason for the shoulder peak of the remaining PEI in the PEEK crystal after extraction. The presence of the shoulder peak before and after extraction qualitatively demonstrated that PEI was not fully extracted. These observations have important effects on the pore structure and the thermodynamic properties of the materials, which will be enhanced by the retention of PEI.

### 3.2. Effect of Phase Structure on Membrane Morphology

The two-phase structure of PEEK/PEI composite directly influences the morphology of PEEK hollow fiber membrane after extraction. The morphologies of the PEEK membranes were investigated as a function of thermal history for the heat treatment of PEEK/PEI composites. The SEM images of the cross-section, internal surface and outer surface of the PEEK membranes are shown in Figure 6, Figure 7 and Figure 8, respectively.

It can be seen that the continuous spongy pore structure formed in the cross-section of the membrane. At the same time, with the increase of the annealing temperature, the pore size and porosity increased in the outer and inner surfaces. This was mainly determined by the solid-liquid phase separation of the PEEK and PEI, which was controlled by the crystallization kinetic of PEEK and the diffusion kinetic of the PEI. When the PEEK/PEI composite was annealed at a low temperature, the kinetic of the phase separation was controlled by the crystallization of PEEK, and PEI chains were trapped primarily in the interlamellar region of the PEEK crystals, which resulted in a small sized pore structure after extraction. However, the diffusion of PEI chains plays a dominant role in phase separation of PEEK/PEI composite when it is annealed at a high temperature, and PEI chains are mainly trapped in interfibrillar or interspherulitic regions, therefore, some large-sized pores can be formed after extraction [24].

It can be seen from Figure 7 and Figure 8 that the pore size and porosity were different at the inner and outer surfaces of the membrane. At the outer surface, the pore sizes were smaller and the porosity was lower than those at the inner surface. This phenomenon is a common phenomenon in the preparation of hollow fiber membrane by TIPS and may be attributed to the different distribution of PEI in PEEK at the inner and outer surfaces of the membrane. Because of the direct contact with ice water, the cooling rate of the outer surface of the membrane was higher than that of the inner surface, therefore, the crystallization of PEEK played a dominant role in the phase separation at the outer surface, resulting in the PEI being primarily trapped in the interlamellar and a dense cortical structure being formed after extraction. However, the phase separation is controlled by the diffusion of PEI chains at the inner surface of the membrane, and most of PEI is outside the crystal of PEEK, leading to the formation of large-sized pore structures with high porosity after extraction [30,31].

### 3.3. Formatting of Mathematical Components

BET-specific surface area measurement is a method used to directly measure the properties and structure of pores by using the amount of adsorbed gas on porous solid materials. According to International Union of Pure and Applied Chemistry (IUPAC), macroporous materials usually contain pores larger than 50 nm in diameter, mesoporous materials are usually defined as porous materials with pore sizes between 2~50 nm and microporous materials are usually defined as porous materials with pore sizes below 2 nm [32]. The nitrogen adsorption isotherm curves of PEEK hollow fiber membranes prepared at different annealing temperatures are shown in Figure 9. According to IUPAC classification, the nitrogen adsorption isotherm curves were of type IV, accompanied by a type H1 hysteresis loop [33]. This result indicates that the porous material obtained from PEEK/PEI composite can be classified as mesoporous. The porous structure is uniform cylindrical, and the pore size distribution in the porous PEEK is uniform.

Nitrogen adsorption BET analysis indicated that the porous PEEK material exhibited uniform pore size distribution. The pore size distribution curves of the PEEK hollow fiber membranes prepared at different annealing temperatures are shown in Figure 10. It can be seen that with the increase of annealing temperature from 240 to 280 °C, the most probable aperture diameter (most likely pore in the membrane) gradually increased, and the pore diameter of the PEEK membrane increased from 10.59 to 37.85 nm with the increase in the annealing temperature, and the surface area decreased from 111.9 to 83.69 m^2^/g, as shown in Table 1. This was mainly because the control of the phase separation process changed from the crystallization of PEEK to the diffusion of PEI. These results are in agreement with the SEM results.

### 3.4. Effect of Phase Structure on Porosity and Transport Properties

Table 1 shows the pore characteristics and permeability of hollow fiber membranes prepared at different annealing temperatures. The wall thicknesses of the PEEK hollow fiber membranes were obtained by the statistical calculation of the cross-section microscope photos of their precursory fibers, as shown in Appendix A. It can be seen that the overall porosity increased from 55.98% to 59.11% as the annealing temperature increased from 240 to 280 °C. This is a further confirmation of what was observed by SEM analysis. As the voluminous macrovoids raised, the membrane structure became puffier and the overall porosity increased. Another important effect was the water permeability increase at the higher annealing temperature. The water flux of the PEEK hollow fiber membrane increased from 1.91 × 10^−2^ to 1.65 × 10^−1^ Lh^−1^ m^−2^ bar^−1^ as the annealing temperature increased from 240 to 270 °C. The membrane prepared at the annealing temperature of 250 °C had a significant enhancement in pore diameter and water permeability, which may be related to the density of the inner and outer surface of the membrane. From the SEM diagram, it can be seen that the inner and outer surfaces of the membranes prepared by annealing at 240 °C were almost free of pores. More pores were distributed on the inner surface of the membranes prepared at the annealing temperature of 250 °C, and there was a small number of pores distributed on the outer surface. The dense cortical structure may be an important reason for the low water flux of the membrane, which was also the reason why the water flux of the membrane prepared at 250 °C was higher than that of the membrane prepared at 240 °C. The structure of the inner and outer surface of the membrane prepared at the annealing temperatures of 260 and 270 °C was similar to that prepared at 250 °C. Therefore, the water flux showed a slight upward trend with an increase of the annealing temperature. The membrane prepared at the annealing temperature of 280 °C had some larger-sized and loose pores, and the hollow fiber membrane was easily damaged during testing, thus the water flux could not be measured.

## 4. Conclusions

In this work, PEEK hollow fiber membranes were successfully prepared from PEEK/PEI composite via the TIPS method. The phase structure evolution of PEEK/PEI composite and its influence on the pore structure and properties of PEEK hollow fiber membrane were studied in detail. It can be concluded that the heat treatment had a significant effect on the phase separation and it was controlled by the crystallization kinetic of PEEK and the diffusion kinetic of PEI. When PEEK/PEI was annealed at a relatively low temperature, the crystallization of PEEK played a dominant role in the phase separation process, and PEI was mainly trapped in the interlamellar region of PEEK crystallization, and, as a result, PEI was difficult to extract and remove, leading to the membrane with the characteristics of some small-sized pores—low porosity, dense structure and low permeability. On the contrary, the diffusion of the PEI dominates the phase separation, and PEI will distribute in the interfibrillar or interspherulitic regions of PEEK crystallization when PEEK/PEI composite is treated in a relatively high temperature; therefore, PEI can be easily extracted and the PEEK membrane with large-sized pores, high porosity, loose structure and high permeability can be obtained. In summary, the structure and properties of the PEEK hollow fiber membrane can be effectively controlled by regulating heat treatment conditions.

## Figures and Tables

**Figure 1 polymers-11-01398-f001:**
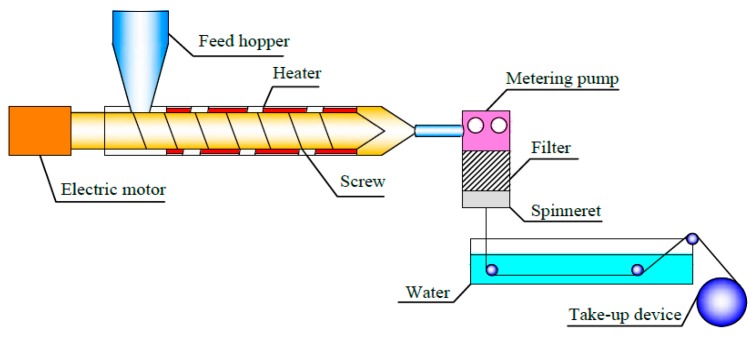
Spinning device diagram of poly(ether ether ketone) (PEEK)/ polyetherimide (PEI) blend hollow fibers.

**Figure 2 polymers-11-01398-f002:**
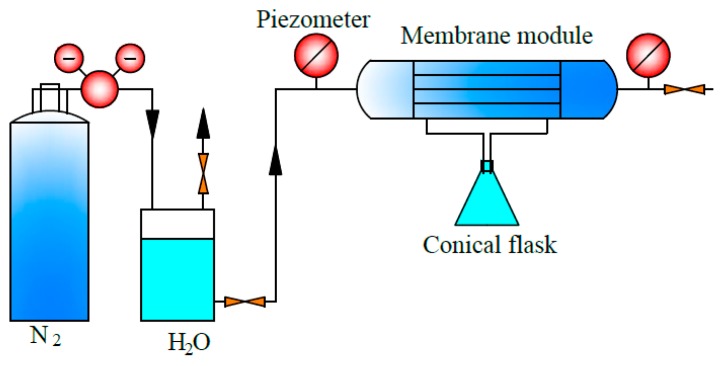
Test device diagram for water flux of the PEEK hollow fiber membrane.

**Figure 3 polymers-11-01398-f003:**
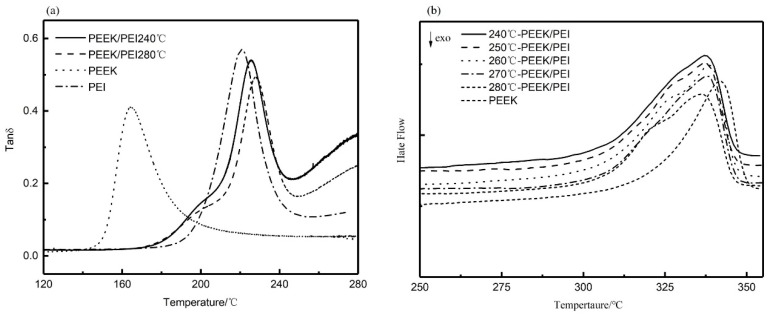
Dynamic mechanical thermal analysis (DMA) (**a**), and differential scanning calorimetry (DSC) (**b**) curves for blend annealing samples.

**Figure 4 polymers-11-01398-f004:**
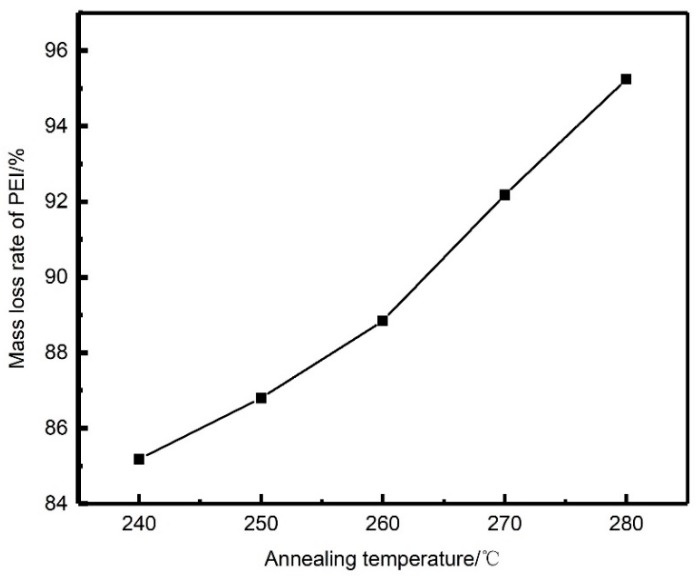
Mass loss rate of PEI after extraction of each annealing sample.

**Figure 5 polymers-11-01398-f005:**
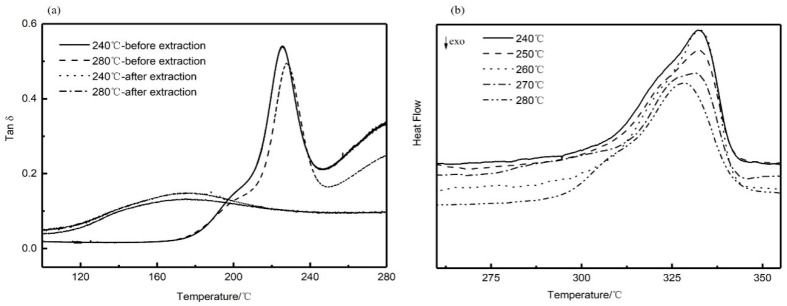
DMA (**a**) and DSC (**b**) curves for blend annealing samples after extraction.

**Figure 6 polymers-11-01398-f006:**
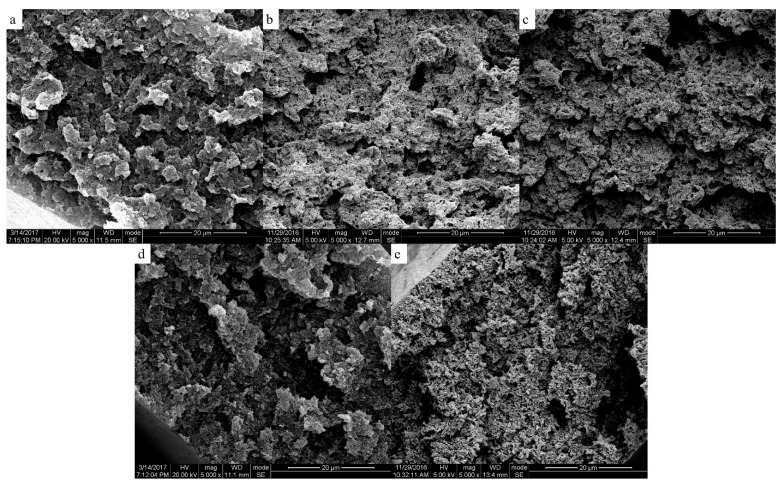
Scanning electron micrographs of the cross-section of the membrane; samples were cold crystallized at (**a**) 240 °C; (**b**) 250 °C; (**c**) 260 °C; (**d**) 270 °C; (**e**) 280 °C.

**Figure 7 polymers-11-01398-f007:**
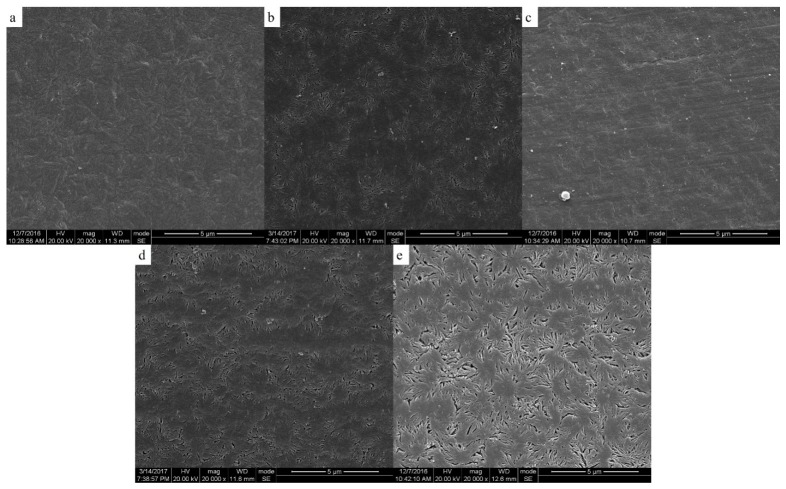
Scanning electron micrographs of the inner surface of membrane; samples were cold crystallized at (**a**) 240 °C; (**b**) 250 °C; (**c**) 260 °C; (**d**) 270 °C; (**e**) 280 °C.

**Figure 8 polymers-11-01398-f008:**
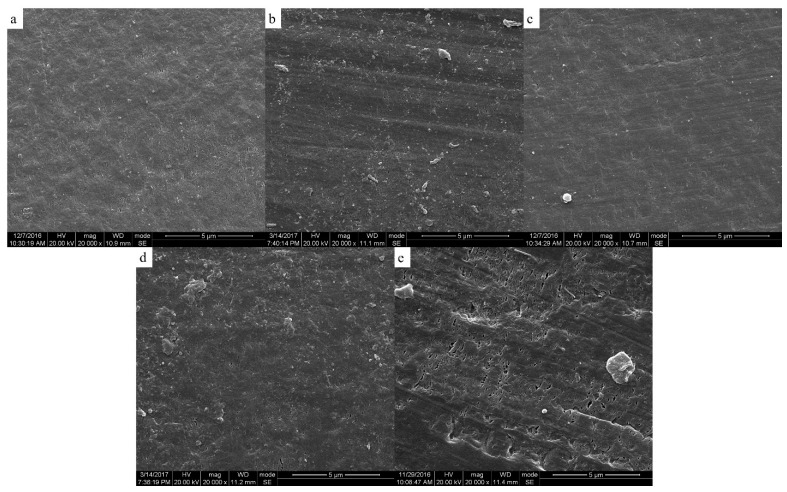
Scanning electron micrographs of the outer surface of membrane; samples wer cold crystallized at (**a**) 240 °C; (**b**) 250 °C; (**c**) 260 °C; (**d**) 270 °C; (**e**) 280 °C.

**Figure 9 polymers-11-01398-f009:**
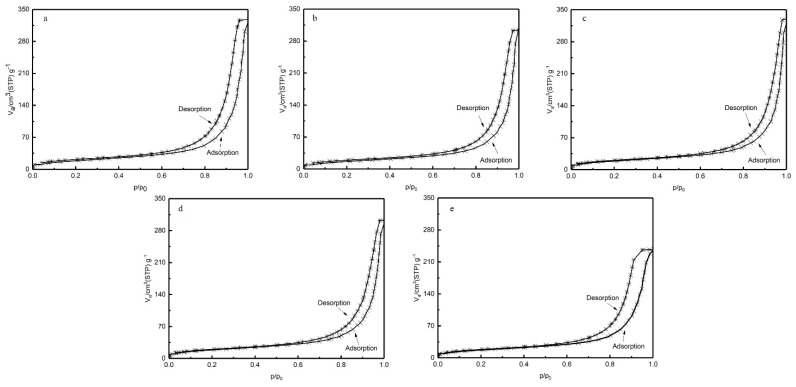
Isotherm of nitrogen adsorption-desorption for the porous PEEK hollow fiber (**a**: 240 °C; **b**: 250 °C; **c**: 260 °C; **d**: 270 °C; **e**: 280 °C).

**Figure 10 polymers-11-01398-f010:**
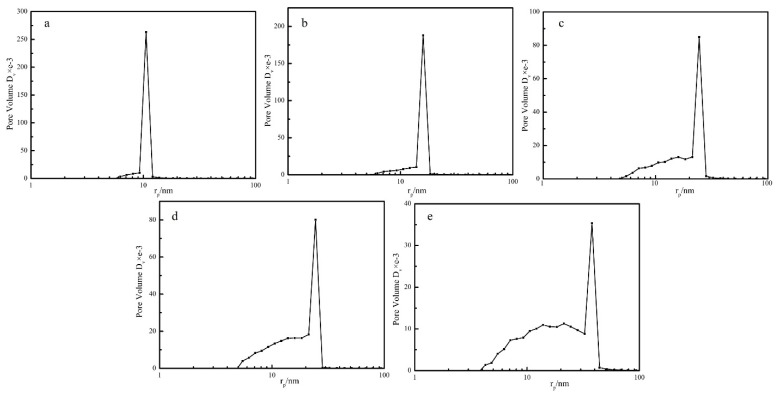
Pore size distribution of PEEK hollow fiber membrane prepared at different annealing temperatures (**a**: 240 °C; **b**: 250 °C; **c**: 260 °C; **d**: 270 °C; **e**: 280 °C).

**Table 1 polymers-11-01398-t001:** Dimension, porosity and water flux of hollow fiber membranes at different annealing temperatures.

Annealing Temperature (°C)	Thickness (µm)	Pore Diameter (nm)	Surface Area (m^2^/g)	Porosity ^1^ (%)	Water Flux (Lh^−1^ m^−2^ bar^−1^)
240	69.63	10.59	111.9	55.98	1.91 × 10^−2^
250		15.94	99.50	56.88	1.30 × 10^−1^
260		24.43	96.13	58.67	1.47 × 10^−1^
270		24.43	83.61	59.56	1.65 × 10^−1^
280		37.85	83.69	59.11	---

^1^ The theoretical porosity is 60.39%; theoretical porosity refers to the porosity of the PEI wholly removed from the PEEK/PEI composite.

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
