# Peer review of "Pore Structure and Properties of PEEK Hollow Fiber Membranes: Influence of the Phase Structure Evolution of PEEK/PEI Composite"

_polymers, 2019, doi:10.3390/polym11091398_

Round 1

Reviewer 1 Report

The manuscript “Pore structure and properties of PEEK hollow fiber membranes: Influence of the phase structure evolution of PEEK/PEI composite” by Chen et al. provides very interesting results on hollow fibers preparation conditions and their structures. The manuscript is well written, the experiments are performed in a correct way; however, I have some questions to the authors: Why the authors submerged the obtained opaque precursory fibers for 9h? Any reference why this specific time is required? Why not 5, 8 or 24h? The authors should provide the fiber size, i.e I am missing a SEM cross section of the fiber, where I could measure its size and the wall size. The authors used the dynamic mechanical thermal analysis to investigate the glass-rubber relaxation characteristics of the polymer blends and to illustrate the nature of PEI phase in the corresponding PEEK/PEI blend. Any comments on different methods to investigate the crystallinity of the fibers? In table 1, the statistic errors should be added.

Reviewer 2 Report

In introduction part: line 31: the authors wrote “…the fluid components avoid phase change….”.  This is not true for all membrane separation process, like membrane distillation, also in pervaporation at desorption step. Please correct this sentence. Please rewrite this sentence: “…makes them cannot be further developed”. In section 2.2: line 93: “PEEK/PEI particles with 40 wt% of PEEK and 60 wt% of PEI weight fraction were prepared by melting extrusion…” Why did the authors choose this particular composition? Please explain. In line 97: Please indicate the temperature instead of “in ice water”. In equation (1), the authors assume the complete removal of PEI. But that is not the case. The overall porosity may be slightly different as the authors did not count the presence of PEI. In section 2.3: line 104-105: “120℃ for 9 h to perform decomposition and removal of PEI from the precursor fibers”. What does this “decomposition” mean? If it decomposes at 120oC, then definitely the decomposition will occur during melt mixing at 360oC. Is the PEEK/PEI a homogeneous blend? The authors wrote in line 179: “…corresponding to the miscible part of PEEK and PEI…” Is there any immiscible part? In Figure 4: Please indicate how did the authors measure this “mass loss rate”. In Figure 6: The cross-sectional SEM images do not show any variation in porosity or pore size across the membrane. Please provide new SEM or correct this part. In Figure 7: What does this “internal surface of the membrane” mean? Please clarify. In line 247: “pore diameter of PEEK membrane is increasing from 10.59 nm to 37.85 nm with…” Did the authors calculate this value or obtained from the measuring instrument? If calculated, please provide details. In Table 1: What does this specific diameter (m2/g) mean? Please define in the text.

Round 2

Reviewer 2 Report

Please change the sentence "fluid components avoid phase change during membrane separation..." as "fluid components avoid phase change during most membrane separation..." and replace the 'most' from "...with most traditional distillation processes." In response to previous comment 3: Please indicate how did the authors control the temperature. In response to previous comment 4: Please indicate this explanation in the revised manuscript. In response to previous comment 7: Please indicate this part in the revised manuscript and also the measuring procedure.
